# New Antifungal Compound: Impact of Cosolvency, Micellization and Complexation on Solubility and Permeability Processes

**DOI:** 10.3390/pharmaceutics13111865

**Published:** 2021-11-04

**Authors:** Tatyana V. Volkova, Olga R. Simonova, German L. Perlovich

**Affiliations:** Gennady Alekseevich Krestov Institute of Solution Chemistry of the Russian Academy of Sciences, 153045 Ivanovo, Russia; vtv@isc-ras.ru (T.V.V.); ors@isc-ras.ru (O.R.S.)

**Keywords:** solubility, permeability, solubilization, micellization, complexation

## Abstract

Poor solubility of new antifungal of 1,2,4-triazole class (S-119)—a structural analogue of fluconazole in aqueous media was estimated. The solubility improvement using different excipients: biopolymers (PEGs, PVP), surfactants (Brij S20, pluronic F-127) and cyclodextrins (α-CD, β-CD, 2-HP-β-CD, 6-O-Maltosyl-β-CD) was assessed in buffer solutions pH 2.0 and pH 7.4. Additionally, 2-HP-β-CD and 6-O-Maltosyl-β-CD were proposed as promising solubilizers for S-119. According to the solubilization capacity and micelle/water partition coefficients in buffer pH 7.4 pluronic F-127 was shown to improve S-119 solubility better than Brij S20. Among biopolymers, the greatest increase in solubility was shown in PVP solutions (pH 7.4) at concentrations above 4 *w*/*v*%. Complex analysis of the driving forces of solubilization, micellization and complexation processes matched the solubility results and suggested pluronic F-127 and 6-O-Maltosyl-β-CD as the most effective solubilizing agents for S-119. The comparison of S-119 diffusion through the cellulose membrane and lipophilic PermeaPad barrier revealed a considerable effect of the lipid layer on the decrease in the permeability coefficient. According to the PermeaPad, S-119 was classified as a highly permeated substance. The addition of 1.5 *w*/*v*% CDs in donor solution moves it to low-medium permeability class.

## 1. Introduction

Fungal infections caused by numerous pathogens are a serious threat to human health, animals, plants etc. Constant reproduction of fungi can lead to the damage of skin and internal organs. Pathogenic fungi actively mutate and become increasingly resistant to the drugs used. They are especially dangerous for people with weakened immunity, in particular, for those who have had a coronavirus infection and who have taken steroids and immunosuppressant for treatment [1].

To date, a vast array of antifungals exists on the market. Among them the triazole class compounds, exemplified by fluconazole, itraconazole, econazole, terconazole, butoconazole, tioconazole, voriconazole, posaconazole, ravuconazole are abundant [2]. These drugs are active as the specific inhibitors of cytochrom P-450 enzyme lanosterol 14-α-demethylase [3], which determines the biosynthesis of ergosterol—the main structural component of the fungus cell membrane [4,5]. The problem with treating fungal infections is quite difficult since it requires the creation of effective and non-toxic drugs acting exclusively against them. Moreover, unfortunately, fungi develop resistance to many drugs used. Such an evolutionary process pushes the medical scientists to a constant searching for new antifungal molecules.

Modern drug discovery methods tend to advance large and hydrophobic molecules, which are likely to suffer from limited solubility and low bioavailability [6,7]. The pharmaceutical industry widely uses the drug delivery systems that can mitigate these risks. To achieve solubility improvement, the systems based on: micronization [8], nanoparticle formation [9], lipid-based formulations [10], cocrystallization [11], formation of host-guest complexes with cyclodextrins [10] and associates with biopolymers [12] and micellar surfactants [13] are widely applied. Estimation of drug solubility in mixtures of water and cosolvents can provide valuable information on solubility enhancement and the factors controlling the solubility phenomenon. The ability of the surfactants to form the micelles in aqueous solution is due to the formation of self-assemblies, which can incorporate poorly soluble drug into the hydrophobic core, while the hydrophilic part provides protection against micelle-protein interactions that contribute to longer-lasting action and stability [13]. Moreover, preventing the precipitation of the drug in solution allows the maintenance of the required therapeutic level of the drug in the blood. In turn, cyclic oligosaccharides—cyclodextrins forming soluble ‘host-guest’ type complexes with drug compound in solution have been successfully applied for drug solubilization, both in the research environment and in clinical use. The ability of cyclodextrin to form an inclusion complex with a guest molecule is determined by the steric factor and the thermodynamic interactions between different components of the system (cyclodextrin, guest, solvent) [14]. Toxicity studies have demonstrated lack of absorption of CDs from the gastrointestinal tract [15]. A vast array of native and modified CDs is applied in pharmaceutics. Amongst them, β-CD is of a high importance since it has a potential for complex formation due to availability, low cost and high efficiency [14] but, at the same time, has several limitations, such as limited aqueous solubility, nephrotoxicity because of the interactions with lipid membrane components [15]. In turn, many branched derivatives are highly soluble and bio-adaptable. Among them, 2-hydroxypropyl-β-cyclodextrin and 6-O-Maltosyl-β-cyclodextrin, which have been used in the present study, possess high aqueous solubility and lower toxicity as compared to β-cyclodextrin [16]. Furthermore, 6-O-Maltosyl-β-cyclodextrin synthesized by enzymatic addition of maltose to the external (primary) hydroxyl groups of the initial β-cyclodextrin macrocycle through an α-(1→6) glycosidic linkage. Being a useful functional excipient in the pharmaceutical industry [17], this glycosylated cyclodextrin does not have a pronounced taste and is suitable for use even in the food industry [14]. Notably, 2-hydroxypropyl-β-cyclodextrin has greater applicability in pharmaceutics than 6-O-Maltosyl-β-cyclodextrin.

The literature data survey contains studies on the solubility enhancement of different antifungals, for example, itraconazole [18,19,20], voriconazole [21], propiconazole [22], posaconazole [23,24,25] via cyclodextrin complex formation and polymers [26,27]. Among them there are studies devoted to fluconazole—structure analogue of S-119—the object of the present investigation. As it has been identified, within the literature, the solubility of fluconazole was improved in β-cyclodextrin [28,29], 2-hydroxypropyl-β-cyclodextrin and sulfobutyl ether-β-cyclodextrin [30] solutions. Besides this, the increased dissolution rate of fluconazole in solid dispersions with β-cyclodextrin, PEG-6000 and PVP K30 was demonstrated [31]. Taking into account the results of our previous study [32] where a poor solubility of S-119 in aqueous solutions was estimated (4.45 × 10^−3^ mol∙L^−1^ at pH 2.0 and 4.01 × 10^−5^ mol∙L^−1^ at pH 7.4) and data from the literature review it seemed reasonable to check the possibility of S-119 solubility improvement via cyclodextrins and polymers.

Since the drug bioavailability, distribution and accumulation in the tissues is determined not only by the solubility but also by the diffusion through the biological membranes, evaluation of the permeability during the drug formulation design is no less important. Namely, the permeability of drugs can reduce in the presence of solubilizing agents and cyclodextrins [33,34] and, as follows, should be evaluated through the study.

This work is aimed to increase the solubility of S-119, which was proposed as a novel potential drug compound for the prevention and treatment of fungi. The compound can be considered as structurally related to well-known antifungal fluconazole. In order to solve the problem of poor solubility of S-119, the solubilization in the aqueous solutions of biopolymers such as polyethylene glycol 6000 (PEG 6000), polyethylene glycol 35000 (PEG 35000), polyvinylpyrrolidone K29-32 (PVP), polyethylene glycol octadecyl ether (Brij S20), triblock copolymer poly(ethylene glycol)-block-poly(propylene glycol)-block-poly(ethylene glycol)-pluronic F127 (F127) and cyclodextrins: α-cyclodextrin (α-CD), β-cyclodextrin (β-CD), 2-hydroxypropyl-β-cyclodextrin (HP-β-CD), and 6-O-Maltosyl-β-cyclodextrin (O-M-β-CD) was studied by phase solubility method at two pH values (pH 2.0 and pH 7.4).

The structures of S-119 and all the excipients used are given in Figure 1.

The influence of the excipient structure on the manifestation of the solubilizing effect was analyzed and discussed in terms of solubilization, micellization and complex formation. The diffusion of S-119 through the synthetic membrane composed of regenerated cellulose and biomimetic membrane PermeaPad barrier was evaluated to reveal the influence of the excipients on the permeability and shed a light on the impact of lipid bilayer on the permeation process.

## 2. Materials and Methods

### 2.1. Materials

Investigated compound S-119.

New hybrid of 1,2,4-triazole derivative 3,3′-(piperazine-1,4-diyl)bis(2-(2,4-difluorophenyl)-1-(1*H*-1,2,4-triazol-1-yl)propan-2-ol) (S-119) was synthesized at the Gause Institute of New Antibiotics (Moscow, Russia). Synthetic procedure applied to the synthesis of S-119 was reported in the previous study [32].

NMR ^1^H NMR (400 MHz, DMSO-d6) δ: 8.27 (d, J = 8.3 Hz, 2H), 7.72 (d, J = 8.8 Hz, 2H), 7.35 (dtd, J = 15.9, 9.0, 6.8 Hz, 2H), 7.11 (dddd, J = 11.9, 8.8, 6.0, 2.6 Hz, 2H), 6.91 (qd, J = 8.6, 2.6 Hz, 2H), 5.61 (s,2 0H), 4.52 (d, J = 2.8 Hz, 4H), 2.79 (q, J = 12.4 Hz, 2H), 2.60–2.56 (m, 2H), 2.40–2.26 (m, 8H).

NMR ^13^C NMR (101 MHz, DMSO-d6) δ: 163.29, 163.16, 160.85, 160.73, 160.63, 160.51, 158.17, 158.05, 150.82, 145.30, 130.18, 130.11, 130.08, 130.01, 126.72, 126.68, 126.59, 126.55, 111.19, 111.17, 110.99, 110.96, 104.39, 104.13, 74.82, 74.77, 63.93, 63.90, 56.09, 56.05, 54.63.

Excipients.

Polyethylene glycol 6000 (M_w_ = 6000 Da) was from Acros Organics (New Jersey, NJ, USA), polyethylene glycol 35000 (M_w_ = 35000 Da), polyvinylpyrrolidone K29-32 (M_w_ = 58,000 Da), Brij S20 (M_w_ = 1152 Da), pluronic F127 (M_w_ = 12,600 Da) and 2-hydroxypropyl-β-cyclodextrin (M_w_ = 1380 Da) were purchased from Sigma-Aldrich (St. Louis, USA). β-Cyclodextrin (M_w_ = 1135 Da), 6-O-maltosyl-β-cyclodextrin (M_w_ = 1452 Da) were obtained from CycloLab.LTD (Budapest, Hungary) and α-cyclodextrin (M_w_ = 973 Da)—from FlukaAnalytical (Steinheim, Germany).

Solvents and reagents.

1-octanol (purity ≥99%), n-hexane (purity ≥ 0.97%), and ethanol (purity 95.0%) were obtained from Sigma-Aldrich (St. Louis, MO, USA).

Buffer components.

Potassium dihydrogen phosphate (purity ≥ 99%), and disodium hydrogen phosphate dodecahydrate (purity ≥ 99%)—from Merk (Darmstadt, Germany); potassium chloride (purity ≥ 99%), and hydrochloric acid 0.1 mol⋅dm^−3^ fixanal—from Aldrich.

Buffer preparation.

Phosphate buffer pH 7.4 (I = 0.15 mol·L^−1^) was prepared from KH_2_PO_4_ (9.1 g in 1 L) and Na_2_HPO_4_·12H_2_O (23.6 g in 1 L). For the preparation of acidic buffer pH 2.0 (I = 0.10 mol·L^−1^), 6.57 g of KCl was dissolved in water and 119.0 mL of 0.1 mol∙L^−1^ hydrochloric acid was added. After this, the volume of the solution was adjusted to 1 L with bidistilled water.

Bidistilled water (electrical conductivity 2.1 μS cm^−1^) was used for buffer solutions preparation. The pH was checked using a pH meter FG2-Kit Mettler Toledo (Greifenzee, Switzerland) standardized with pH 1.68, 6.86 and 9.22 solutions.

All the materials for the experiments were used as received.

### 2.2. Methods

#### 2.2.1. Phase Solubility Study

The effect of different excipients on the solubility of S-119 was studied according to the phase solubility method [35] at 25.0 ± 0.1 °C. Excess amounts of S-119 were added to buffered aqueous solutions containing increasing concentrations of excipients. Suspensions were mixed in a thermostat at 25 °C for 72 h to attain equilibrium, centrifuged (Biofuge pico, Thermo Electron LED GmbH (Langenselbold, Germany) at 9000 rpm for 20 min. Concentrations of the saturated solutions of S-119 were determined spectrophotometrically (Shimadzu 1800 (Kyoto, Japan) at λ = 261 nm. The experiments were performed in triplicate with the accuracy of 2–4%. The phase solubility diagrams were obtained by plotting the amount of dissolved compound (M) versus the amount of excipient added (M).

The stability constant ($K_{S}$) was calculated using the framework of Higuchi and Connors model [35]:(1)$$K_{S}=\frac{slope}{S_{2}^{0}(1-slope)}$$
where *slope* is calculated from of the linear phase solubility diagram, and $S_{2}^{0}$ is the equilibrium concentration of S-119 in pure buffer (pH 2.0 or pH 7.4) at 25.0 ± 0.1 °C.

#### 2.2.2. In Vitro Permeability Assay

Permeability experiments were performed using the reverse dialysis set-up adapted to the Franz diffusion cell (PermeGear, Inc., Hellertown, PA, USA). This method was successfully applied by di Cagno et al. [36] for the permeability evaluation of a whole set of diverse drugs. The donor solution of the investigated compound was placed in the lower chamber. The membrane was mounted between the donor and acceptor chambers. At the start of the experiment, the upper (acceptor) compartment was filled with the respective pure buffer. The donor solution was stirred with a magnetic stirrer bar. The samples of 0.5 mL were withdrawn from the receptor solution each 30 min over 5 h and replaced with the same volume of pure buffer. The volume of the donor solution in the lower compartment was 7 mL and the receptor volume was 1 mL. A schematic diagram of the equipment is illustrated in Figure 2.

Two types of barriers were applied for the permeability evaluation: the regenerated cellulose membrane (RC) with a molecular weight cut off MWCO 12–14 kDa (Standard Grade RC Dialysis Membrane, Flat Width 45 mm) and the PermeaPad (PP) barrier (PHABIOC, Germany, www.permeapad.com, accessed on 20 May 2021). The PermeaPad barrier was used as received from the supplier. The cellulose membrane was pretreated with distilled water for 30 min and the traces of water were removed with filter paper before use. The membrane effective surface area was 0.785 cm^2^. Buffer solution pH 2.0 was applied throughout the experiment due to impossibility of the drug detection in the receptor solution at pH 7.4, where S-119 solubility is extremely low. The permeability coefficients of S-119 were measured with buffer pH 2.0 and 1.5 *w*/*v*% of α-CD, HP-β-CD and O-M-β-CD in the donor compartment. The temperature of the system was maintained at 37.0 ± 0.1 °C. The concentrations of the sample solutions were measured using spectrophotometer (Spectramax 190; Molecular devices, Molecular Devices Corporation, California, CA, USA) in 96-well UV black plates (Costar) at λ = 261 nm. The permeation plots were constructed as the amount of the permeated drug over the surface area (dQ/A) versus time (t). Flux (*J*) was calculated as a slope of permeation plots by the equation:(2)$$J=\frac{dQ}{A\times dt}$$

The slope (*J*) normalized by the concentration of the drug in the donor compartment (*C*_0_) represents the apparent permeability coefficient (*P_app_*):(3)$$P_{app}=\frac{J}{C_{0}}$$

The average value of *P_app_* from at least 3 experiments was taken into consideration.

Sink conditions were maintained at any time meaning that the drug concentration in the acceptor chamber did not exceed 10% of the drug concentration in the donor chamber.

## 3. Results

### 3.1. Phase Solubility Study

As it was shown in our previous study [32] by PXRD experiment, S-119 is a solid crystalline substance. It has poor solubility in aqueous media (4.45·10^−3^ M and 4.01·10^−5^ M in buffers pH 2.0 and pH 7.4, respectively, at 25 °C), which would be a reason for its inadequate transport properties and bioavailability. In the present work, we made attempts to improve the solubility with the help of different pharmaceutical excipients (solubilizing agents). To this end, the equilibrium solubility of S-119 was measured at several concentrations of polymers (including those forming the micelles in solution) and cyclodextrins. The concentrations of the saturated solutions were determined by UV-spectroscopy. The spectra of the solutions in the presence of all excipients were similar to those in pure buffers at both pHs (λ = 261 and 266 nm) indicating no changes in S-119 state. As an example, the plots of UV spectra for the solutions in buffer pH 7.4 are introduced in Appendix A (Appendix A). As previously shown [32], the solubility of the substance is affected by the pH of aqueous medium. Due to this fact, the experiments were carried out in two pHs in order to reveal the simultaneously impact of the pH and excipient on the solubility. The solubility of the compound at different concentrations of the excipients is listed in Appendix A. As follows from the table, a pronounced effect of pH on the shape of the solubility dependences was revealed. An increase in S-119 apparent solubility in all systems was observed at the elevated solubilizers concentrations in buffer pH 7.4, as opposed to buffer pH 2.0, where a diverse solubility behavior in the presence of excipients was revealed.

#### 3.1.1. Solubilization in Cyclodextrins Solutions

Figure 3 illustrates the phase solubility diagrams for S-119 in CD solutions.

It seems interesting to compare the degree of the solubility enhancement by CD complexation of the newly investigated drug (S-119) and the closest structural analogue—fluconazole. According to the literature [28], only a 1.1-fold solubility growth was obtained for fluconazole in 0.0016 M aqueous β-CD solution as compared to 6.8-fold for S-119 (0.0015 M β-CD, pH 7.4). Analogous regularity is characteristic for HP-β-CD—the solubilization is more effective for S-119 (a 24-fold increase in 0.01 M CD) as compared to a 1.2-fold solubility increase in 0.004 M HP-β-CD for fluconazole. Interestingly, Fernández-Ferreiro et al. [30] reported a 4-fold higher solubility of fluconazole in 0.08 M aqueous HP-β-CD, as compared to our data for S-119, which indicates a 151-fold solubility growth in 0.07 M CD. A further study by Kırımlıoğlu et al. [29] revealed 1.84-fold fluconazole solubility enhancement in aqueous 0.02 M β-CD. Bearing in mind that we use buffer solution at pH 7.4 and the experiments by authors [28,29,30] deal with unbuffered water, the difference can be attributed both to the structural features of the studied compounds and the ionization state of the molecules. In addition, the observed situation clearly proved the validity of using buffer solutions when ionizable compounds are under study.

The solubilizing potential of cyclodextrins towards S-119 antifungal was evaluated using the stability constants (*K_S_*) of the complexes derived from the phase solubility profiles (Figure 3) calculated by Equation (1). According to the Higuchi and Connors classification [35], at both pHs, the phase solubility diagrams for α-CD, HP-β-CD and O-M-β-CD are of *A_L_*-type, indicating the formation of soluble complexes. As opposed, the *B_S_*-type diagrams with linear region from 0 to 0.003 M CD concentration (Figure 3, inserts) were obtained for β-CD. The observed regularity is commonly encountered in case of-CD complex formation. It can be explained by the formation of supramolecular complexes of high molecular weight with limited solubility as a rule precipitating in solutions of naturally occurring CDs, especially β-CD [37]. An insoluble complex was precipitated, causing turbidity of the solvent and lowering of the compound concentration. The slope values of the phase-solubility diagrams are lower than unity, indicating the 1:1 stoichiometry of the complexes in the investigated range of cyclodextrin concentrations [21]. The calculated *K_S_* values of the complexes are illustrated in Figure 4 and listed in Appendix A along with the complexation efficiency and drug:CD molar ratio.

Evidently, neutral S-119 (at pH 7.4) exhibits higher affinity to CDs than cationic particles (at pH 2.0), due to an essential contribution of the hydrophobic effect on complex formation [38]. Importantly, only the stability constants obtained by S-119 with all CDs at pH 7.4 belong to the range from 200 to 5000 M^−1^. As stated by Szejtli [39], this range meets a sufficient potential bioavailability due to an optimal drug release and amount of free drug intended to be absorbed in the lipid layer of the biological membrane. In turn, the stability of the complexes forming in acidic solution (pH 2.0) is low (from 5 to 27.4 M^−1^ for α- and β-CD, respectively), which is coming from a small number of intermolecular interactions between the CDs and compound. As such, this can promote a rapid release of a substance and reduce the influence of complexation on the solubility [39]. The following sequence of the stability constants was stated: α-CD < HP-β-CD < O-M-β-CD < β-CD (Appendix A, Figure 4) at both pHs. A similar trend was reported for fluconazole/HP-β-CD and fluconazole/β-CD complexes by Orgovan et al. [40]: the stability constants for neutral state of the FCZ molecules with β- and HP-β-CD as 93.3 M^−1^ and 43.6 M^−1^, respectively, were derived by the authors from the phase solubility diagrams. Interestingly, the complexes of S-119 at pH 7.4 are essentially more stable (47- and 52-fold for β- and HP-β-CD, respectively), as compared to those of FCZ. Undoubtedly, the lipophilicity of the compounds plays a crucial role in the interactions with the hydrophobic CD cavity. According to log*D^oct^* = 0.5 [41], FCZ belongs to the substances of moderate lipophilicity, whereas for S-119 log*D^oct^* = 1.43 [32]. The *K_S_* values for the cationic species of FCZ in acidic solution were shown to be lower (10.7 and 13.5 M^−1^), similarly to S-119, according to a poor affinity of the charged particles to the hydrophobic cavity of CD [42]. This trend is correlated with the intrinsic solubility $S_{2}^{0}$, which is significantly lower for the uncharged S-119 molecules. Obviously, a driving force for the inclusion in the CD cavity is lower when the aqueous solubility is higher [43]. The stability constants of FCZ in unbuffered water (68.7 M^−1^ and 34.6 M^−1^ for β- and HP-β-CD, respectively) were reported by Fernández-Ferreiro et al. [30] and follow the regularity *K_S_* (β-CD) > *K_S_* (HP-β-CD). The more stable complexes of nimodipine with β-CD as compared to HP-β-CD were obtained by Kopecky et al. [44]. The possible reason for the observed regularity is hindering the penetration of a bulky compound molecule into the cavity by the hydroxypropyl group [45] and, thus, lowering the complexation ability in comparison to the parent CD, especially at the higher degree of substitution. The least stable complex with α-CD as compared to parent β-CD and derivatives occurs probably due to the geometrically more suitable cavity of β-CDs for incorporation of the aromatic fragment of the S-119 molecule. 

Moreover, the molecular cavity of α-CD is too small, and hinders a deep insertion of the bulky S-119 molecule. A somewhat lower stability was estimated for S-119/O-M-β-CD complex as compared to S-119/β-CD. The small difference indicates only negligible steric hindrance of maltose residue attached to the primary hydroxyl group of β-CD. In a whole, maltosyl branches only minimally affect the inclusion complex formation compared to native β-CD. In spite of the higher stability of S-119/β-CD complexes, undoubtedly, branched HP-β-CD and O-M-β-CD are better solubilizers for poorly water-soluble S-119, since their high aqueous solubility allows for the achievement of much higher concentrations of the compound and no solid complexes precipitate even at higher CD concentrations.

#### 3.1.2. Solubilization by Biopolymers

The phase solubility diagrams of S-119 in polymers solutions at pH 2.0 and pH 7.4 are illustrated in Figure 5.

A pronounced effect of pH on the solubility in different polymer solutions was estimated. A linear dependence of the solubility in the presence of both PEGs and PVP is observed in acidic solution (Figure 5a) with the following range of polymers in accordance with the solubility enhancement at 10 *w*/*v*% polymer concentration: PEG 6000 (1.45-fold) > PVP (1.32-fold) > PEG 35000 (1.20-fold). As follows, the increase in S-119 solubility is rather low and very close for all polymers used.

Notably, non-linear solubility dependences were obtained for S-119 at pH 7.4 (Figure 5b). Only a slight increase in solubility up to 4 *w*/*v*% polymer concentrations was achieved. At the same time, a dramatic solubility growth above 4 *w*/*v*% takes place. In these systems the dissolution at low excipient concentrations is associated with the predominance of the processes strengthening the structure of water by the hydrophobic mechanism, since upon the small additions of polymers their non-polar groups stabilize the structure of water [46]. In turn, the polar groups of the excipients can replace the water molecules in the nodes of the ice-like framework and destroy the quasicrystalline structure of water by forming hydrogen bonds. The cumulative influence of all these factors determines the compound solubility at low contents of a non-aqueous component [47]. The dramatic solubility increase, from 4 to about 7 *w*/*v*% of the excipients, is due to the processes of destruction of quazicrystalline structure of water with polymer molecules and replacement of water molecules in the solvate shell of S-119 by the polymeric molecules which promotes the process of dissolution. Moreover, in buffer solution at pH 7.4, the consequence of the polymers in respect of the extent of the solubilization changed: PVP (232-fold) > PEG 6000 (193-fold) > PEG 35000 (79-fold). Highly probably, the reason for this phenomenon is similar to CD solutions, namely, a driving force for the association between the compound and polymer is lower when the aqueous solubility is higher. Taken together, the presence of the non-polar regions in the polymer structures can reduce the water polarity and weaken its intermolecular hydrogen-bonding network, resulting in a solubility increase in a non-polar substance. At the same time, for ionized (at pH 2.0) species of S-119, this effect is not pronounced due to the affinity to hydrophilic aqueous medium. In addition, the ability of polymers to change the hydration state of the substance, per se, and to form water-soluble associates with the excipients [48], via van der Waals interactions and hydrogen bonding, are also responsible for the solubility increase [49]. For the sake of comparison, the reported data [50] on the solubility enhancement of imidazole class antifungal—clotrimazole show 17.5-fold solubility growth in 20% solution of PVP K30, which is essentially lower than for S-119 in 20% PVP (232-fold).

#### 3.1.3. Solubilization with Non-Ionic Surfactants

Brij S20 and F-127 used in the present study belong to the non-ionic surfactants which are considered as favorable due to a low toxicity and effective solubilizing power under biorelevant conditions [51]. The dependences of S-119 solubility on Brij S20 and F-127 concentrations are plotted in Figure 5. As follows from Figure 5, even a certain decrease in S-119 solubility in both surfactants solutions at pH 2.0 is observed. Obviously, the ionized particles of the compound do not undergo the incorporation into the micelles of Brij S20 and F-127. As it was reported [52], the more nonpolar the solute, the more likely it is to be incorporated in the core of the micelle. Moreover, the increase in the surfactant concentration can reduce the number of hydrogen binding sites available for the association of the compound and surfactant. At that point, the surfactant molecules interact with water, affect its structuring and hinder the interactions with positively charged species. As opposed to pH 2.0, at pH 7.4 a pronounced growth of S-119 solubility (199-fold and 281-fold for Brij S20 and F-127, respectively) in 10 *w*/*v*% was detected. The solubilizing potential of the micelles of both surfactants towards S-119 in solution from the linear phase solubility diagrams at pH 7.4 was obtained. To this end, the solubilization capacity (χ) representing the amount of the compound that can be solubilized by one mole of polymer was calculated using the following equation [53]:(4)$$\chi =(S_{2}-S_{2}^{0})/(C_{surf}-CMC)$$
where $S_{2}^{}$ and $S_{2}^{0}$ are the total solute solubility at a particular concentration of surfactant solution and the intrinsic solute solubility, respectively; *C_surf_* is the surfactant concentration; *CMC* is the surfactant critical micelle concentration. All concentrations are expressed in molarity. Taking into account that above the CMC the concentration of surfactant monomers is equal to the CMC, the concentration of micelles can be determined as (*C_surf_* − *CMC*). The *CMC* of Brij S20 and F127 were taken from the literature [54,55,56,57]. From this, the plot of $\left(S_{2}-S_{2}^{0}\right)$ on (*C_surf_* − *CMC*) dependence was used for the solubilization capacity determination (Appendix A for S-119/Brij S20 and S-119/F-127). From Appendix A, a higher solubilization capacity of F127 (1.42 ± 0.01) than that of Brij S20 (0.092 ± 0.003) with respect to S-119 was estimated. Besides this, a micelle/water molar partition coefficient (*K_m_*_/*w*_) was derived using the following equation [58]:(5)$$K_{m/w}=S_{m}/S_{2}^{0}=(S_{2}-S_{2}^{0})/S_{2}^{0}$$

The molar partition coefficient [52] reflects the potential ability of the micelle systems to act as drug carriers and shows a ratio between the drug concentration in the micelle (*S_m_*) and water $\left(S_{2}^{0}\right)$ pseudo phases (similarly to organic solvent/water partition). It can be determined from the slope of the dependence *S_m_*/*S^0^* on surfactant concentration (Appendix A). As a result of the regression, *K_m_*_/*w*_ equal to 2303 ± 29 and 35343 ± 302 for S-119/Brij S20 and S-119/F-127, respectively, were determined, clearly demonstrating a higher solubilizing effect of F127 on S-119. Possibly, availability of extended polypropylenoxide (hydrophobic) block in the structure of F-127 molecule promotes the inclusion of S-119 hydrophobic molecules into the micelles. The obtained results proved that the structure of surfactant strongly influences the solubilizing effect towards a poor soluble substance. This observation is confirmed by the literature data [59] on another antifungal—ketoconazole—for which different solubilizing capacity (χ) (0.15, 0.26 and 0.29), as well as partition coefficient micelle/water (26,779, 33,458 and 35,317) were obtained in such non-ionic surfactants as Myrj 52, Tween 80 and Brij 35, respectively. Evidently, the solubilizing capacity of all these agents with respect to ketoconazole drug is comparable with that of Brij S20 and lower than that of F-127 with respect to S-119. Obviously, the ionization of S-119 species at pH 2.0 makes it absolutely impossible the incorporation of the compound in the micelles of both block copolymers.

### 3.2. Permeability Study

The selection of an in vitro model for the passive drug permeability evaluation is important in the early stage of drug design. In the present study, the transport of S-119 through the regenerated cellulose (RC) membrane and the PermeaPad barrier (PP) was investigated and compared. The PermeaPad barrier is composed of lipid substrate approximating it to the real membranes of the intestine epithelium [36]. As reported [60], semi-permeable cellulose membranes are highly permeable to many drugs and are useful for the optimal design of drug-cyclodextrin formulations. As opposed to lipophilic biological membranes and also the PermeaPad barrier, both drug and drug-CD complexes can diffuse through the membrane of sufficient MWCO [61], for example, 12–14 kDa. As a result, the transport across the RC is influenced by the possibility of drug penetration not only as free molecules but also in the content of CD complexes preventing the precipitation of poorly soluble compound. Since the size of the complex is essentially greater than that of the free drug molecule, its diffusion coefficient is lower, which would decrease the movement through the membrane. From all the aforesaid, rather different modes of permeability across PP and RC membranes should be considered for the interpretation of the results.

The extremely low solubility of S-119 in buffer pH 7.4 makes it impossible to detect the concentrations of the sample solutions taken from the receptor compartment of the Franz cell. As follows, the permeation experiments were carried out at pH 2.0 for all studied systems, as reported in our previous study [32]. The initial experimental data used for the *P_app_* calculation, and the resultant cumulative amounts of the permeated S-119 and FCZ (for comparison), steady state fluxes and permeability coefficients through the PP barrier and RC membrane are given in Table 1. Figure 6a depicts the kinetic dependences of the cumulative amount of the permeated S-119 from pure buffer and from 1.5 *w*/*v*% solutions of α-CD, HP-β-CD and 6-O-M-CD.

As evident from Figure 6a, for pure buffer and all CD solutions, the flux *J* (slope of the kinetic dependence) through the RC membrane is higher than the PP barrier, possibly due both to the high retention of a lipid layer in the PP and the diffusion of both drug and drug/CD complexes through RC (in case of CD solutions). The same regularity of the permeability coefficients is illustrated in Figure 6b. The comparison of the results for S-119 and FCZ revealed an essentially faster diffusion of FCZ through both barriers (1.8- and 1.2-fold for RC and PP, respectively). An essentially larger value of *P_app_* for FCZ through RC as compared to S-119 can be explained by its higher diffusion coefficient in water. As stated by Avdeef et al. [62], diffusion coefficients in pure water (*D_w_*) at room temperature can be calculated by the empirical formula:log*D_w_* = −4.113 − 0.4609 log*MW*(6)
where *MW* is the molecular weight of the diffusing molecule. Expectedly, the calculations showed the diffusion coefficients equal to 5.51 × 10^−6^ and 4.17 ×10^−6^ for FCZ and S-119, respectively. Interestingly, the difference in the *P_app_* values between these compounds is smoothed in the case of PP barrier. Obviously, the lipophilicity plays a crucial role upon the permeation through a lipophilic layer of PP: a highly lipophilic S-119 (log*D^oct^* = 1.43) more readily passes the lipid layer than moderate lipophilic FCZ (log*D^oct^* = 0.5). Since the permeation coefficient is directly proportional to the diffusion coefficient and lipophilicity [63], the opposite action of these two factors determines the permeability.

The impact of CDs on the permeability of S-119 is illustrated in Figure 6b. Evidently, 1.5 *w*/*v*% CD concentrations in donor solution reduce the permeation through both membranes mainly due to the decrease in the fraction of free (unbounded with CD) drug molecules in case of PP and the permeation of bulky drug/CD complexes across the RC membrane. Expectedly, no strict regularity in the permeability reduction between the PP and RC barriers for CD solutions was detected due to different flexibility of CD molecules on the solution/membrane boundary surface and different mechanisms of permeation. A more pronounced permeability decrease from HP-β-CD (1.87-fold) and O-M-β-CD (2.63-fold) solution was estimated with RC and PP barriers, respectively.

According to the permeability classification proposed by di Cagno et al. [36] for permeability assay based on the PermeaPad barrier, S-119 belongs to highly permeated substances, even in the case of buffer pH 2.0 where the molecule is ionized. As follows, even higher permeability can be proposed for neutral species. At the same time, the presence of CDs in 1.5 *w*/*v*% concentration transfers the compound to medium-low class.

### 3.3. Solubilization, Micelle Formation and Complexation Processes: Thermodynamic Considerations

The favorable/unfavorable character of any process can be assessed using Gibbs free energy. In order to analyze the information on solubilization, micelle formation and complexation obtained in the present study, the standard Gibbs energy (at 25 °C) of the outlined processes was calculated by the following equations:(7)$$\Delta G_{S}^{0}(C_{exc})=-RT\ln\frac{S_{2}^{}}{S_{2}^{0}},$$
where $\Delta G_{S}^{0}(C_{exc})$—the standard free energy of solubilization—transferring a compound from the aqueous buffer solution to the excipient solution of a defined concentration (*C_exc_*), $S_{2}^{0}$ and $S_{2}^{}$—the solubility of a drug in pure buffer and at a defined excipient concentration, respectively; *R* is the universal gas constant; *T* is the standard temperature of 25 °C
(8)$$\Delta G_{C^{}}^{0}=-RT\ln K_{s}^{}$$
where $\Delta G_{C}^{0}$ is the change of the complexation Gibbs energy, *K_S_* is the stability constant of the complexation reaction
(9)$$\Delta G_{m/w}^{0}=-RT\ln K_{m/w}^{}$$
where $\Delta G_{m/w}^{0}$ is the standard Gibbs energy of the partition between the micelle and water phases, *K_m_*_/*w*_ is the partition coefficient.

The values of the Gibbs energies calculated by Equastions (7)–(9) are listed in Appendix A. The negative values of $\Delta G_{S}^{0}$ (except S-119/Brij S20 and S-119/F-127 systems in buffer solution pH 2.0) indicate the spontaneous character of the considered processes. In case of exceptions positive $\Delta G_{S}^{0}$ comes from the decrease in S-119 solubility in the presence of these excipients (See Section 3.1.3). Upon the increase in the excipient concentration, the Gibbs energy increases negatively, thus, facilitating more favorable solubilization. More negative $\Delta G_{S}^{0}$ values were obtained in buffer pH 7.4 as compared to pH 2.0, since the neutral species (at pH 7.4) are readily solubilized and enter the hydrophobic surfactant micelle core or CD cavity. The comparison of these parameters in buffer pH 7.4 at 7 *w*/*v*% of the excipients revealed very close $\Delta G_{S}^{0}$ between PEG 35000 (−12.23 kJ∙mol^−1^) and F-127 (−13.06 kJ∙mol^−1^) with the exceptions of α-CD ($\Delta G_{S}^{0}$ = −7.73 kJ∙mol^−1^) and β-CD for which the concentration of 7 *w*/*v*% cannot be prepared due to a low β-CD solubility. The driving force of the solubilization in acidic medium has maximal negative values for O-M-β-CD (−1.49 kJ∙mol^−1^) and HP-β-CD (−1.27 kJ∙mol^−1^).

Comparative analysis of $\Delta G_{}^{0}$ showed the prevalence of the driving force of micellization and complexation over solubilization. The maximal potential of F-127 as micelle forming agent ($\Delta G_{m/w}^{0}$ = −25.96 kJ∙mol^−1^) and O-M-β-CD as an effective means of complexation ($\Delta G_{C}^{0}$ = −20.36 kJ∙mol^−1^) confirmed the results of the S-119 solubility improvement by these excipients. Expectedly, the minimal $\Delta G_{C}^{0}$ = −4.00 kJ∙mol^−1^ and $\Delta G_{C}^{0}$ = −14.17 kJ∙mol^−1^ for α-CD in pH 2.0 and pH 7.4, respectively, was derived in full agreement with the solubility results. Notably, among cyclodextrins the highest values of $\Delta G_{C}^{0}$ = −8.21 kJ∙mol^−1^ (pH 2.0) and $\Delta G_{C}^{0}$ = −20.81 kJ∙mol^−1^ (pH 7.4) accompanied by rather low $\Delta G_{C}^{0}$ parameters were determined for β-CD, which once more proved that the potential of β-CD as a solubilizing agent for S-119 is restricted, due to its poor aqueous solubility and tendency to form insoluble complexes in solution.

## 4. Conclusions

In the present study the effect of biopolymers: Polyethylene glycol 6000 (PEG 6000), polyethylene glycol 35000 (PEG 35000), polyvinylpyrrolidone K29-32 (PVP); non-ionic surfactants: Polyethylene glycol octadecyl ether (Brij S20), triblock copolymer poly(ethylene glycol)-block-poly(propylene glycol)-block-poly(ethylene glycol)—pluronic F127 (F127); and cyclodextrins: α-cyclodextrin (α-CD), β-cyclodextrin (β-CD), 2-hydroxypropyl-β-cyclodextrin (HP-β-CD) and 6-O-Maltosyl-β-cyclodextrin (O-M-β-CD) on the solubility of new antifungal (S-119) at pH 2.0 and pH 7.4 was investigated. A pronounced effect of pH on the solubility in excipients solutions was estimated. The extremely low values of the stability constants of the ionized species of S-119 in acidic CDs solutions were revealed. As opposed, at pH 7.4 the stability constants with all CDs belong to the optimal range of good potential bioavailability. HP-β-CD and O-M-β-CD were proved to be better solubilizers for poorly soluble S-119, since their high aqueous solubility allows for achievement of much higher concentrations of the compound, and no solid complexes precipitate even at higher CD concentrations. A considerable S-119 solubility growth in buffer solution at pH 7.4 following the order: PVP (232-fold) > PEG 6000 (193-fold) > PEG 35000 (79-fold) was estimated. A pronounced increase of S-119 solubility at pH 7.4 (199-fold and 281-fold for Brij S20 and F-127, respectively) in 10 *w*/*v*% was detected, as opposed to pH 2.0 where the solubility is even lowered. A higher solubilizing effect of F127 as compared to Brij S20 towards S-119 was revealed from solubilization capacity and micelle/water molar partition coefficient. Complex analysis of the driving forces of the solubilization, micellization and complexation processes proved the solubility results and suggested pluronic F-127 and O-maltosyl-β-CD as promising solubilizing agents for poor soluble S-119 compound. The comparison of the S-119 permeability through the regenerated cellulose membrane and lipophilic PermeaPad barrier revealed a considerable effect of the lipid layer on the decrease in the permeability coefficient. According to the PermeaPad, S-119 was classified as highly permeated substance. The addition of 1.5 *w*/*v*% CDs in donor solution moves it to low-medium permeability class.

## Figures and Tables

**Figure 1 pharmaceutics-13-01865-f001:**
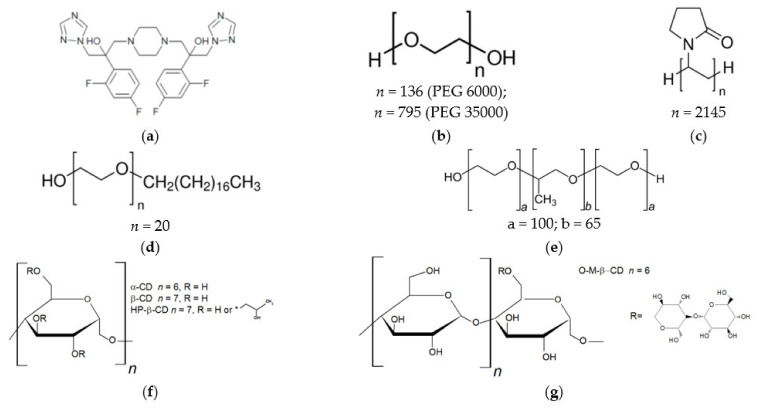
Structure of the studied compound S-119 (**a**), and the excipients: PEG (**b**); PVP (**c**); Brij S20 (**d**); F127 (**e**); α-CD, β-CD, HP-β-CD (**f**) and O-M-β-CD (**g**).

**Figure 2 pharmaceutics-13-01865-f002:**
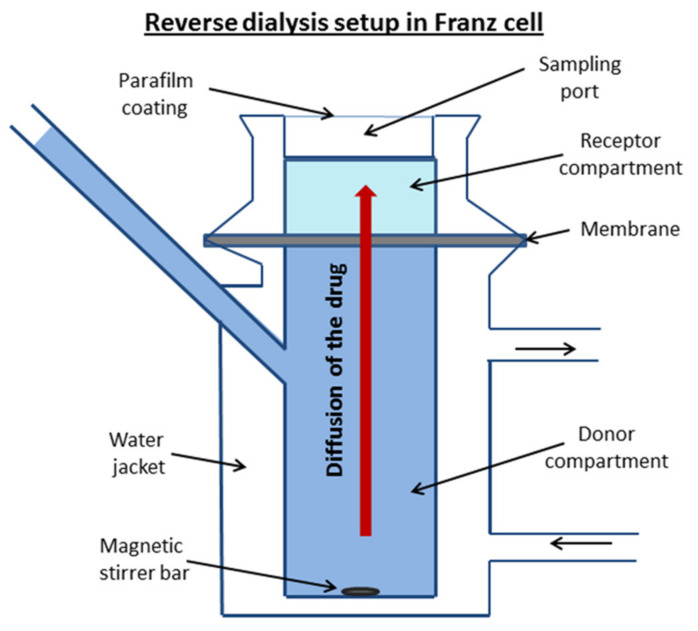
A schematic diagram of the reverse dialysis set-up adapted to the Franz diffusion cell.

**Figure 3 pharmaceutics-13-01865-f003:**
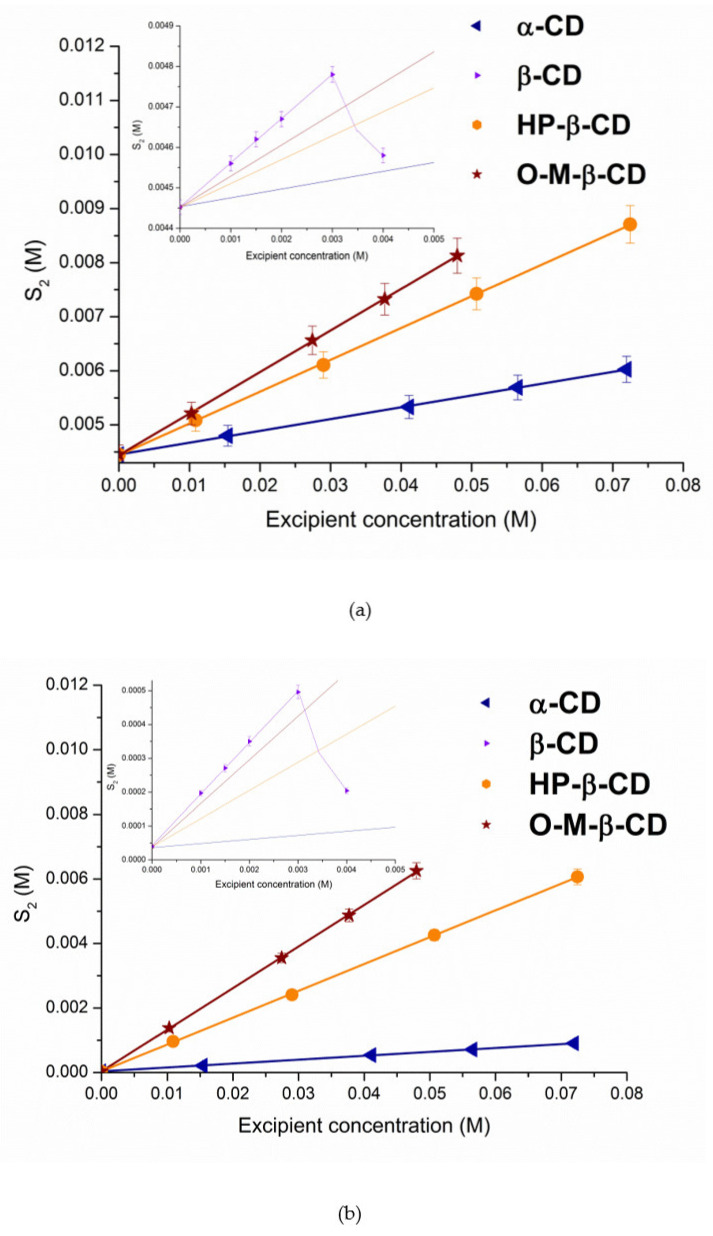
Solubility of S-119 in solutions of different concentrations of cyclodextrins: (**a**) pH 2.0 and (**b**) pH 7.4 (◄—α-CD, ►—β-CD, ●—HP-β-CD, ★—O-M-β-CD) at 25.0 ± 0.1 °C.

**Figure 4 pharmaceutics-13-01865-f004:**
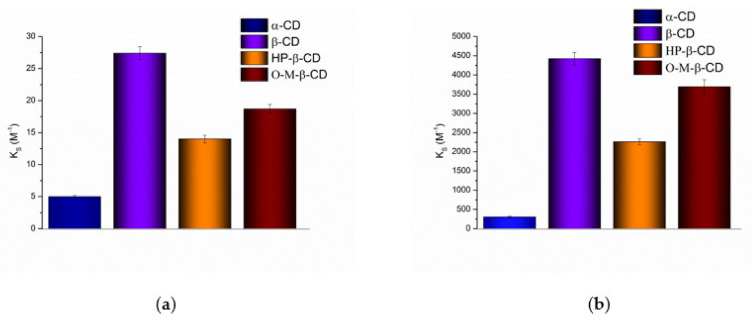
Stability constants (*K_S_*) of S-119/CD complexes in buffer pH 2.0 (**a**) and buffer pH 7.4 (**b**) at 25.0 ± 0.1 °C.

**Figure 5 pharmaceutics-13-01865-f005:**
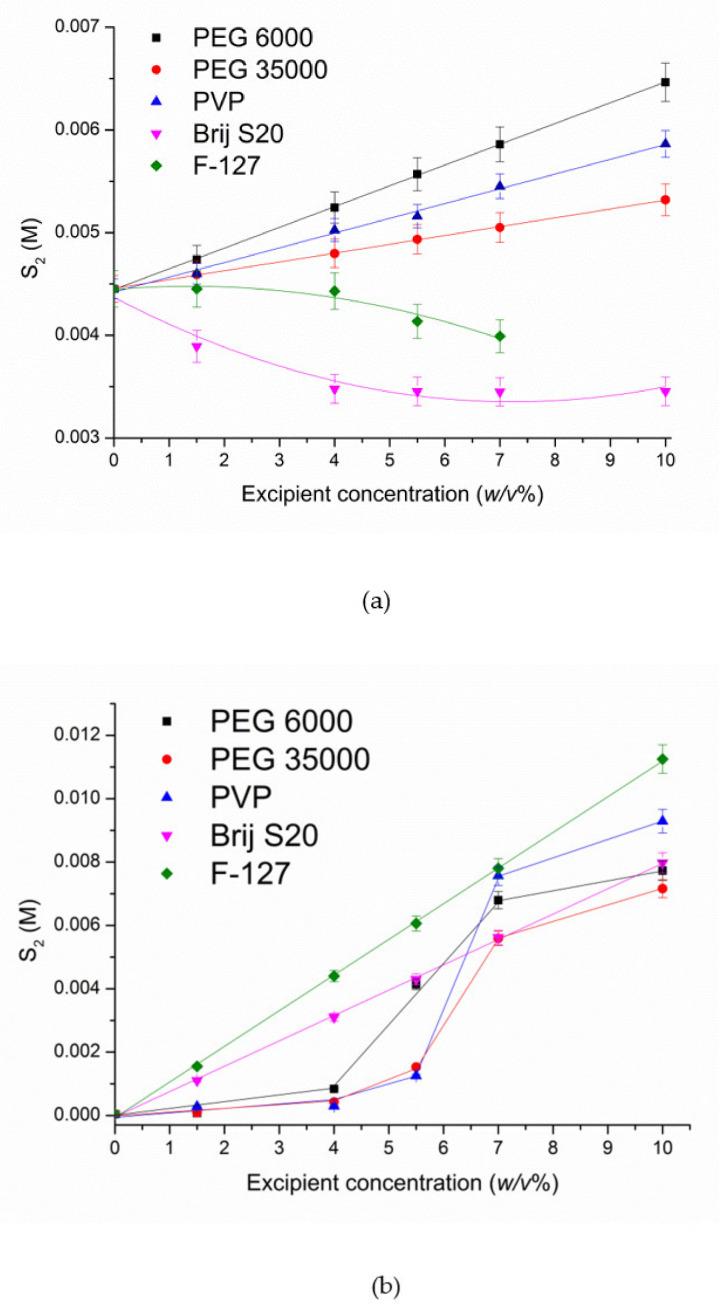
Solubility of S-119 in solutions of different concentrations of polymers: (**a**) pH 2.0 and (**b**) pH 7.4 (■—PEG 6000, ●—PEG 35000, ▲—PVP, ▼—Brij S20, ♦—F-127) at 25.0 ± 0.1 °C.

**Figure 6 pharmaceutics-13-01865-f006:**
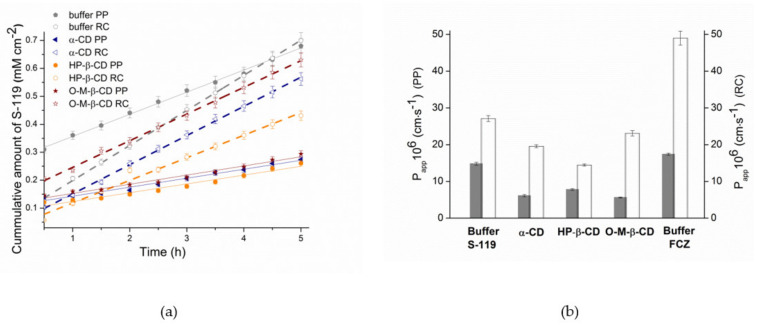
S-119 permeability through the RC membrane (empty symbols, dotted lines) and PP barrier (filled symbols, straight lines) in the presence of 1.5% *w*/*v* cyclodextrins: (**a**) S-119 transport (amount of the cumulative amount permeated vs. time); (**b**) permeability coefficients (*P_app_*) at 37 °C: grey columns—permeability coefficients through the PermeaPad barrier; white columns—values of the permeability coefficients through the cellulose membrane. Data presented as mean ± S.D.; *n* = 3.

**Table 1 pharmaceutics-13-01865-t001:** Donor solution concentrations (C), steady penetrate rate—flux (J), and permeability coefficients (*P*_app_) of S-119 and FCZ in pure buffer pH 2.0 and in 1.5 *w*/*v*% CD solutions.

System	PermeaPad (PP)			Cellulose Membrane (RC)		
	C (M)	J (µM∙cm^−2^∙s^−1^)	*P*_app_ (cm∙s^−1^)	C (M)	J (µM∙cm^−2^∙s^−1^)	*P*_app_ (cm∙s^−1^)
	S-119					
Buffer	1.50 × 10^−3^	2.20 × 10^−5^	(1.48 ± 0.04) × 10^−5^	1.29 × 10^−3^	3.49 × 10^−5^	(2.71 ± 0.08) × 10^−5^
β-CD	1.63 × 10^−3^	9.93 × 10^−6^	(6.12 ± 0.31) × 10^−6^	1.49 × 10^−3^	2.90 × 10^−5^	(1.96 ± 0.04) × 10^−5^
HP-β-CD	1.28 × 10^−3^	1.00 × 10^−5^	(7.81 ± 0.24) × 10^−6^	1.53 × 10^−3^	2.21 × 10^−5^	(1.45 ± 0.03) × 10^−5^
6-O-M-β-CD	1.81 × 10^−3^	1.01 × 10^−5^	(5.62 ± 0.14) × 10^−6^	1.16 × 10^−3^	2.68 × 10^−5^	(2.31 ± 0.07) × 10^−5^
	FCZ					
Buffer	2.01 × 10^−3^	3.50 × 10^−5^	(1.74 ± 0.03) × 10^−5^ *	2.67 × 10^−3^	1.31 × 10^−4^	(4.90 ± 0.19) × 10^−5^

* —taken from [31].

## Data Availability

The results obtained for all experiments performed are shown in the manuscript and SI, the raw data will be provided upon request.

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
