# Peer review of "New Antifungal Compound: Impact of Cosolvency, Micellization and Complexation on Solubility and Permeability Processes"

_pharmaceutics, 2021, doi:10.3390/pharmaceutics13111865_

Round 1

Reviewer 1 Report

The article “New antifungal compound: Impact of cosolvency, micellization and complexation on solubility and permeability processes” is an interesting work that brings into discussion the solubility problems of a new antifungal compound of 1,2,4-triazole class, this property being important to achieve desired concentration of drug in systemic circulation in order to obtain a desired pharmacological response. The authors used in this study the effect of a large number of biopolymers on the solubility of new antifungal drug, the whole study being well explained and very well argued from a scientific point of view. The literature used is from recent years indicating that the authors have been guided by the latest studies and tried to bring to the readers' attention the new achievements that appeared in this field. In my opinion, this paper is well-written and might be interesting and useful for researchers in this field and I recommend the paper publication in Pharmaceutics journal.

However, I have the following suggestion: a brief discussion regarding the comparison of S-119 solubility enhancement with other similar studies found in the literature can be added.

Author Response

Response to Reviewer 1

The article “New antifungal compound: Impact of cosolvency, micellization and complexation on solubility and permeability processes” is an interesting work that brings into discussion the solubility problems of a new antifungal compound of 1,2,4-triazole class, this property being important to achieve desired concentration of drug in systemic circulation in order to obtain a desired pharmacological response. The authors used in this study the effect of a large number of biopolymers on the solubility of new antifungal drug, the whole study being well explained and very well argued from a scientific point of view. The literature used is from recent years indicating that the authors have been guided by the latest studies and tried to bring to the readers' attention the new achievements that appeared in this field. In my opinion, this paper is well-written and might be interesting and useful for researchers in this field and I recommend the paper publication in Pharmaceutics journal.

However, I have the following suggestion: a brief discussion regarding the comparison of S-119 solubility enhancement with other similar studies found in the literature can be added.

Response

A brief discussion regarding the comparison of S-119 solubility enhancement with the respective references from some other similar studies has been added in the Results and Discussion Section.

Reviewer 2 Report

The manuscript of Volkova et al presents an interesting study of the encapsulation and permeability of a fungal using differents agents. They prepared a comparison between differents materials and they study the permeation of the fungal using 2 membrane systems. In my opinion, the manuscript could be interesting for readers.

In general terms the manuscript is quite interesting and well presented. The results are well described. However, some points needs to be clarified.

  • L72. A reference is desirable.
  • Figure 1. Some images are from well known chemical suppliers, I suggest to use creative commons images.
  • Figure 2. Could the authors explain the decrease of solubility with bCD?
  • All images. The bar errors should be displayed.
  • Discussion. Several references are needed when authors explain results. Please check.
  • Section 3.2. The cellulose membrane cutoff let CDs and severals materials to be released too. In my opinion, the real effect of CDs cannot be measured in this way. I recommend to use a 1KDa membrane or similar.

Author Response

Response to Reviewer 2

The manuscript of Volkova et al presents an interesting study of the encapsulation and permeability of a fungal using differents agents. They prepared a comparison between differents materials and they study the permeation of the fungal using 2 membrane systems. In my opinion, the manuscript could be interesting for readers.

In general terms the manuscript is quite interesting and well presented. The results are well described. However, some points needs to be clarified.

Answer

L72. A reference is desirable.

Response

The appropriate reference has been introduced.

Answer

Figure 1. Some images are from well-known chemical suppliers, I suggest to use creative commons images.

Response

The images have been reconstructed as creative commons images.

Answer

Figure 2. Could the authors explain the decrease of solubility with bCD?

Response

The observed regularity (decrease of solubility with b-CD) is commonly encountered in case of b-CD complex formation. It can be explained by the formation of supramolecular complexes of high molecular weight with limited solubility as a rule precipitating in solutions of naturally occurring CDs, especially β-CD (Brewster and Loftsson, Adv. Drug Deliv. Rev. 59(7), 645–666).

The necessary explanation and reference have been added to the manuscript (Section 3.1.1.).

Answer

All images. The bar errors should be displayed.

Response

The bar errors have been displayed.

Answer

Discussion. Several references are needed when authors explain results. Please check.

Response

The text of the Results (Section 3) has been checked. The missing references have been introduced.

Answer

Section 3.2. The cellulose membrane cutoff let CDs and severals materials to be released too. In my opinion, the real effect of CDs cannot be measured in this way. I recommend to use a 1KDa membrane or similar.

Response

Thank you for valuable comment. In our work we followed some literature reports on the permeability evaluation in the presence of solubilizing agents using the semi-permeable membranes (Ex.: Masson et al. Jurnal of Controlled Release 59 (1999) 107–118; Loftsson et al. Int. J. Pharm. 232 (2002) 35–43; Ng et al. Pharmaceutics 2 (2010) 209–223). We agree that the partly transition of the drug complexed with CD causes some discrepancies in the interpretation of the results. We will take it into account in our future investigations.

Reviewer 3 Report

The research content involved in this manuscript is relatively simple and lacks significant novelty. This manuscript is not suitable for publication in this journal at present.

Author Response

Response to Reviewer 3

This manuscript studied the solubility and permeability of S-119. Different excipients were tested for solubility and permeability improvement. The authors clearly explained the mechanism back of the solubility and permeability impact factors. In general, this manuscript described a valuable topic with good quality. However, there are some questions about experiment design and result presentation. Please check the following comments.

Major comments:

Answer

For in vitro permeability assay method (section 2.2.2). The author motioned “The cellulose membrane was pretreated with distilled water for 30 min and 161 dried under air before use” (line 161, page4). Why the author chose distilled water not the testing buffer for pre-treatment? Why air dry the membrane before permeation study. Commonly, membrane applied in vertical diffusion cells was immersed in release (permeation) medium 30min and applied for test without trying process. Please explain it. In addition, was there any membrane validation experiment conducted before the permeability study?

Response

Thank you very much for valuable recommendation. The cellulose membrane was pretreated with distilled water for 30 min to remove the preservative and then rinsed in distilled water according to the recommendations of the supplier. After wetting, the traces of water were removed with filter paper before use (not specially dried). The respective clarification of the membrane pretreating has been introduced in Section 2.2.2. At the same time the pretreating with the respective buffer solution, according to your recommendations, seems us reasonable to employ in our future work.

The validity of both membranes was proved in the study of Wu et al. European Journal of Pharmaceutical Sciences (2019), https://doi.org/10.1016/j.ejps.2019.105026. Moreover, the applicability of the PermeaPad barrier for the permeability evaluation in a wide range of pH (including pH 2.0 and pH 7.4) and in the presence of different cosolvents was reported by Bibi et al. (International Journal of Pharmaceutics, 2015, 493(1-2), 192–197). The usability of cellulose membranes for the investigation of cyclodextrin solutions was shown by Loftsson et al. (Int J Pharm. 2002, 232(1-2), 35–43).

On our part, we carried out the permeation experiment on a reference substance Caffeine and compare the results with the study of Di Cagno et al. (Eur. J. Pharm. Sci. 2015, 73, 29–34). The experiments showed good agreement within the limits of experimental error.

Minor comments:

Answer

There is no error bar in Figure 2, 3, 4, and 5.

Response

Error bars have been displayed in Figures 2, 3, 4, and 5.

Answer

In figure 5b, what are the gray and white column represented?

Response

In figure 5b the grey columns and the white columns mean the values of the permeability coefficients through the PermeaPad barrier and through the cellulose membrane, respectively. This explanation has been added in the figure caption.

Answer

For permeability study, why only conducted the study at pH2.0? Is there any specific reason for no study under pH7.4?

Response

A single cause of the permeability study at pH 2.0 (not pH 7.4) is impossibility of detecting the concentrations of S-119 samples taken from the receptor solution due to an extremely low solubility of S-119 at pH 7.4. In our previous study on S-119 solubility and permeability in pure solvents (Volkova et al., J. Mol. Liq. 2021, 336, 116535) the permeability of structurally related fluconazole was evaluated using the PermeaPad barrier at both pH 2.0 and pH 7.4 and the comparative analysis has been carried out to the aim of disclosing the pH impact on the permeability.

   Of course, usually the studies in the region of pH 7.4 or pH 6.8 are performed in order to evaluate the permeability through the intestinal epithelium membranes.

The respective explanation has been introduced in the manuscript (Section 3.2.).

Reviewer 4 Report

This manuscript studied the solubility and permeability of S-119. Different excipients were tested for solubility and permeability improvement. The authors clearly explained the mechanism back of the solubility and permeability impact factors. In general, this manuscript described a valuable topic with good quality. However, there are some questions about experiment design and result presentation. Please check the following comments.  

Major comments:

  1. For in vitro permeability assay method (section 2.2.2). The author motioned “The cellulose membrane was pretreated with distilled water for 30 min and 161 dried under air before use” (line 161, page4). Why the author chose distilled water not the testing buffer for pre-treatment? Why air dry the membrane before permeation study. Commonly, membrane applied in vertical diffusion cells was immersed in release (permeation) medium 30min and applied for test without trying process. Please explain it. In addition, was there any membrane validation experiment conducted before the permeability study?

Minor comments:

  1. There is no error bar in Figure 2, 3, 4, and 5.
  2. In figure 5b, what are the gray and white column represented?
  3. For permeability study, why only conducted the study at pH2.0? Is there any specific reason for no study under pH7.4?

Author Response

Response to Reviewer 4

This manuscript studied the solubility and permeability of S-119. Different excipients were tested for solubility and permeability improvement. The authors clearly explained the mechanism back of the solubility and permeability impact factors. In general, this manuscript described a valuable topic with good quality. However, there are some questions about experiment design and result presentation. Please check the following comments.

Major comments:

Answer

For in vitro permeability assay method (section 2.2.2). The author motioned “The cellulose membrane was pretreated with distilled water for 30 min and 161 dried under air before use” (line 161, page4). Why the author chose distilled water not the testing buffer for pre-treatment? Why air dry the membrane before permeation study. Commonly, membrane applied in vertical diffusion cells was immersed in release (permeation) medium 30min and applied for test without trying process. Please explain it. In addition, was there any membrane validation experiment conducted before the permeability study?

Response

Thank you very much for valuable recommendation. The cellulose membrane was pretreated with distilled water for 30 min to remove the preservative and then rinsed in distilled water according to the recommendations of the supplier. After wetting, the traces of water were removed with filter paper before use (not specially dried). The respective clarification of the membrane pretreating has been introduced in Section 2.2.2. At the same time the pretreating with the respective buffer solution, according to your recommendations, seems us reasonable to employ in our future work.

The validity of both membranes was proved in the study of Wu et al. European Journal of Pharmaceutical Sciences (2019), https://doi.org/10.1016/j.ejps.2019.105026. Moreover, the applicability of the PermeaPad barrier for the permeability evaluation in a wide range of pH (including pH 2.0 and pH 7.4) and in the presence of different cosolvents was reported by Bibi et al. (International Journal of Pharmaceutics, 2015, 493(1-2), 192–197). The usability of cellulose membranes for the investigation of cyclodextrin solutions was shown by Loftsson et al. (Int J Pharm. 2002, 232(1-2), 35–43).

On our part, we carried out the permeation experiment on a reference substance Caffeine and compare the results with the study of Di Cagno et al. (Eur. J. Pharm. Sci. 2015, 73, 29–34). The experiments showed good agreement within the limits of experimental error.

Minor comments:

Answer

There is no error bar in Figure 2, 3, 4, and 5.

Response

Error bars have been displayed in Figures 2, 3, 4, and 5.

Answer

In figure 5b, what are the gray and white column represented?

Response

In figure 5b the grey columns and the white columns mean the values of the permeability coefficients through the PermeaPad barrier and through the cellulose membrane, respectively. This explanation has been added in the figure caption.

Answer

For permeability study, why only conducted the study at pH2.0? Is there any specific reason for no study under pH7.4?

Response

A single cause of the permeability study at pH 2.0 (not pH 7.4) is impossibility of detecting the concentrations of S-119 samples taken from the receptor solution due to an extremely low solubility of S-119 at pH 7.4. In our previous study on S-119 solubility and permeability in pure solvents (Volkova et al., J. Mol. Liq. 2021, 336, 116535) the permeability of structurally related fluconazole was evaluated using the PermeaPad barrier at both pH 2.0 and pH 7.4 and the comparative analysis has been carried out to the aim of disclosing the pH impact on the permeability.

   Of course, usually the studies in the region of pH 7.4 or pH 6.8 are performed in order to evaluate the permeability through the intestinal epithelium membranes.

The respective explanation has been introduced in the manuscript (Section 3.2.).

Round 2

Reviewer 2 Report

The authors have improved the manuscript, but some minor revisions still need.

  1. References 14 and 17 are quite old, please update them and introduction.
  2. An experiment to verify the hypothesis is desirable. Are some solid parts formed? Maybe to quantify the unsoluble mass or to check the UV spectra of the drug (Some changes could be justify by aggregation formation).
  3. About section 3.2, If authors agreed about the possibility of <<partly transition of the drug complexed with CD causes some discrepancies in the interpretation of the results>>, they should modify the manuscript according to this idea.

Author Response

Response to Reviewer 2

Answer

The authors have improved the manuscript, but some minor revisions still need.

1) References 14 and 17 are quite old, please update them and introduction.

Response

References 14 and 17 have been updated (red color)

Answer

2) An experiment to verify the hypothesis is desirable. Are some solid parts formed? Maybe to quantify the unsoluble mass or to check the UV spectra of the drug (Some changes could be justify by aggregation formation).

Response

As it was evidenced while measuring the solubility, no any changes in the UV spectra were detected (two absorption peaks at 261 nm and 266 nm). As an example, the desirable plots of UV spectra for the solutions in buffer pH 7.4 have been added to Supporting Information as Fig. S1.

Answer

3) About section 3.2, If authors agreed about the possibility of <<partly transition of the drug complexed with CD causes some discrepancies in the interpretation of the results>>, they should modify the manuscript according to this idea.

Response

Thank you for highly valuable recommendation. We have introduced the discussion concerning the drug transition in the presence of CDs. The text of the manuscript has been modified in accordance with the comments (red color).

Reviewer 3 Report

This manuscript evaluates the effects of different excipients on the solubility and permeability of a new poor soluble antifungal drug (S-119). Although it may lack some novelty, this study is very interesting and enlightening and it also has certain reference significance for the study of solubilization or permeability improvement of other similar drugs. The quality of the revised manuscript has been greatly improved and can be considered for acceptance and publication after minor revision.

1. As for the method of “In vitro permeability assay” in the line 168-169, is the donor solution in the lower compartment? What is the direction of drug permeation? Please provide a schematic diagram of the equipment for drug in vitro permeability assay so that readers can better understand your experimental design.

2. Drug crystal form also has an important influence on the solubility of drugs. Please introduce the physical properties of S-119, for example, is it solid or oily? Does it have a fixed crystal form?

3. What is the effect of temperature on the solubility of S-119? What is the reason for choosing 25℃ in the study of drug solubility? Why not choose the same temperature as the permeability assay (37℃)?

4. Please analyze the reason for the better solubility of the drug at low pH. Can the compound salified with other acid? The authors believe that, due to the extremely low solubility of S-119 at pH 7.4, the permeation experiments were only carried out at pH 2.0. However, the sample solution in the permeability assay has excipient solubilization, why the drug concentration is still too low to be detected? Please explain it.

Author Response

Response to Reviewer 3

Answer

This manuscript evaluates the effects of different excipients on the solubility and permeability of a new poor soluble antifungal drug (S-119). Although it may lack some novelty, this study is very interesting and enlightening and it also has certain reference significance for the study of solubilization or permeability improvement of other similar drugs. The quality of the revised manuscript has been greatly improved and can be considered for acceptance and publication after minor revision.

  1. As for the method of “In vitro permeability assay” in the line 168-169, is the donor solution in the lower compartment? What is the direction of drug permeation? Please provide a schematic diagram of the equipment for drug in vitro permeability assay so that readers can better understand your experimental design.

Response

In our permeation experiments we use the so-called reverse dialysis set up adapted to the Franz diffusion cell. This method was successfully applied by di Cagno et al. (di Cagno et al. Eur. J. Pharm. Sci. 2015, 73, 29–34.) for the permeability evaluation of a whole set of diverse drugs. The donor solution of the investigated compound was placed in the lower chamber. The membrane was mounted between the donor and acceptor chambers. At the start of the experiment the upper (acceptor) compartment was filled with the respective pure buffer. The donor solution was stirred with a magnetic stirrer. The samples of 0.5 mL were withdrawn from the receptor solution each 30 min over 5 hours and replaced with the same volume of pure buffer.

 The explanation of the method and a schematic diagram of the equipment for the drug in vitro permeability assay have been added in the manuscript Section 2.2.2. for better understanding of the readers (red color)

Answer

  1. Drug crystal form also has an important influence on the solubility of drugs. Please introduce the physical properties of S-119, for example, is it solid or oily? Does it have a fixed crystal form?

Response

The solid crystalline state of S-119 was shown in our previous study (Volkova et al. J. Mol. Liq. 2021, 336, 116535), S-119 by PXRD experiment. The respective information and reference have been added in the manuscript Section 3.1. (red color)

Answer

  1. What is the effect of temperature on the solubility of S-119? What is the reason for choosing 25C in the study of drug solubility? Why not choose the same temperature as the permeability assay (37C)?

Response

The effect of the temperature on S-119 solubility was studied by us before (Volkova et al. J. Mol. Liq. 2021, 336, 116535). An irrelevant difference in the solubility at 25C and 37C was estimated: 1.5-fold in buffer pH 2.0 and 1.2-fold in pH 7.4.

We use 25℃ for the solubility experiments as a standard temperature, whereas, 37℃ in the permeation studies was taken as the temperature very close to 36.6℃ of the healthy humans. These discrepancies in the temperatures didn't complicate the interpretation of the results.

At the same time, we agree that it would be better to take 37℃ for both experiments and we'll take into account in our further studies.

Answer

  1. Please analyze the reason for the better solubility of the drug at low pH. Can the compound salified with other acid?

Response

As it was reported before in our work on the solubility of S-119 in pure solvents (Volkova et al. J. Mol. Liq. 2021, 336, 116535), a weak basic nature is characteristic for S-119 similar to fluconazole due to the proton affinity of both piperazine and triazole ring nitrogens. The weak basic properties explain the pH-dependent solubility of the investigated compound.

Answer

The authors believe that, due to the extremely low solubility of S-119 at pH 7.4, the permeation experiments were only carried out at pH 2.0. However, the sample solution in the permeability assay has excipient solubilization, why the drug concentration is still too low to be detected? Please explain it.

Response

The problem of the determination of the permeability of extremely poor soluble substance consists in impossibility of the detection of the drug concentrations in the receptor solution after the transfer through the membrane. The concentration of the compound in the receptor solution at each time point of the experiment should not exceed 10% of the donor solution concentration (in-sink conditions).

The presence of the excipients doesn't adjust the situation since in most cases the excipients reduce the permeability, thus, decreasing the concentration in the receptor solution.

Round 3

Reviewer 2 Report

The authors have solved the reamining questions with the exception of Supplementary figure 1.

  • It is solubility of the drug in presence of the excipients? The caption should be more clear.
  • The desire experiments should be at a fixed concentration of drug, add more CD and make the spectra to check possible differences in shape when the drug start to be insoluble. Could the authors kindly add the spectra of Figure 3? There are some changes in the shape of the spectra?

After these minor comments, the paper is ready for acceptance if editor consider.

Author Response

Answer

The authors have solved the reamining questions with the exception of Supplementary figure 1.

- It is solubility of the drug in presence of the excipients? The caption should be more clear.

- The desire experiments should be at a fixed concentration of drug, add more CD and make the spectra to check possible differences in shape when the drug start to be insoluble. Could the authors kindly add the spectra of Figure 3? There are some changes in the shape of the spectra?

After these minor comments, the paper is ready for acceptance if editor consider.

Response

- Figure S1 illustrates the spectra of the drug solutions in pure buffer and in the presence of the excipients. The caption to Figure S1 has been rewritten in more clear form.

- We didn't carry out the experiments at fixed drug concentration. In our study we followed the phase solubility analysis of the effect of different agents on the ability of the compound to be solubilized (Higuchi and Connons). An excess of a drug was introduced into several vials filled with cyclodextrin solutions of several concentrations. As it was emphasized by Brewster and Loftsson (M.E. Brewster, T. Loftsson, Adv. Drug Deliv. Rev. 59 (2007) 645–666) "the need for excess drug is based on the desired to maintain as high thermodynamic activity of the drug as possible".

We agree with the Referee that it would be valuable to study the spectral characteristics of a drug under the addition of higher CD concentrations. But considering the perspectives of pharmaceutical formulations (for which the rule "to use as little cyclodextrin as possible in pharmaceutical preparations") based on the investigated compound the addition of large amounts of CDs is inappropriate and was not the aim of the present study.

The spectra of Figure 3 (all cyclodextrins including two spectra of b-CD for the elevated S-119 concentration at 0.001 M b-CD, and for the decreased S-119 concentration 0.004 M b-CD) have been added in Figure S1 (SI). No changes in the shape of the spectra were observed.
